# Synapse elimination activates a coordinated homeostatic presynaptic response in an autaptic circuit

Cecilia D. Velasco [1,2] & Artur Llobet [1,2 ✉]

The number of synapses present in a neuronal circuit is not fixed. Neurons must compensate for changes in connectivity caused by synaptic pruning, learning processes or pathological conditions through the constant adjustment of the baseline level of neurotransmission. Here, we show that cholinergic neurons grown in an autaptic circuit in the absence of glia sense the loss of half of their synaptic contacts triggered by exposure to peptide p4.2, a C-terminal fragment of SPARC. Synaptic elimination is driven by a reorganization of the periodic F-actin cytoskeleton present along neurites, and occurs without altering the density of postsynaptic receptors. Neurons recover baseline neurotransmission through a homeostatic presynaptic response that consists of the coordinated activation of rapid synapse formation and an overall potentiation of presynaptic calcium influx. These results demonstrate that neurons establishing autaptic connections continuously sense and adjust their synaptic output by tweaking the number of functional contacts and neurotransmitter release probability.

[1] Department of Pathology and Experimental Therapy, Faculty of Medicine and Health Science, Institute of Neurosciences, University of Barcelona, 08907 L'Hospitalet de Llobregat, Barcelona, Spain. [2] Laboratory of Neurobiology, Bellvitge Biomedical Research Institute (IDIBELL), 08907 L'Hospitalet de Llobregat, Barcelona, Spain. ✉email: allobet@ub.edu

Synapses must adapt their strength to a constantly changing environment. As a result, synaptic transmission is finely tuned through gain and loss of function mechanisms, which allow the correct function of neuronal circuits[1]. The adaptive properties of synapses can, however, be altered by conditions that modify basal neurotransmission. Changes in excitability, post-synaptic receptor expression, or neurotransmitter release need to be compensated by homeostatic mechanisms that actively return synaptic transmission to the baseline level in order to maintain the stability of neuronal networks[2–4]. The correction of transient unbalances of basal neurotransmission must also take into account that the number of synapses of a given neuronal circuit is not fixed. Synaptic connectivity varies as a function of the developmental stage, learning processes, or certain disease conditions, implying that homeostatic mechanisms have to somehow be able to account for changes in the functional number of contacts present in a given neuronal circuit[5].

During postnatal development there is a pruning of synapses formed exuberantly during embryogenesis, therefore, the correct processing of information must concur with a massive synapse loss. So far it is unclear how neurons adjust a correct number of connections and what are the mechanisms used to balance formation and elimination[6]. This is in part due to the incomplete understanding of mechanisms causing postnatal elimination of synapses[7]. Despite the determinant role of neuronal activity[8], there are growing evidences invoking the participation of glial cells[9]. For example, astrocytes appear to cooperate with the complement system to phagocytose selected synapses[10] or secrete ATP to decrease synapse numbers[11]. Microglia, Schwann cells, or satellite cells also participate in the refinement of synaptic connectivity in the central and peripheral nervous system[9,12]. These evidences together illustrate the important role of neuron–glia interactions in the pruning of neuronal connectivity and bring forward the possible implications of their deregulation.

Among the many molecules released by glia, secreted protein acidic and rich in cysteine (SPARC) emerges as an important cue capable of decreasing synapse numbers[13]. In the nervous system SPARC is exclusively produced by glial cells[14]. It is found enriched in the peripheral processes of astrocytes[15] and its production peaks postnatally[14]. Hence, the enhanced secretion of SPARC could contribute to the developmental pruning of synaptic contacts, as well as, to the elimination of synapses found in certain pathological conditions.

The current work investigates how a neuron establishing an autaptic circuit reacts to the elimination of connectivity driven by SPARC. Although autapses are normally established by certain excitatory and inhibitory neurons[16–18], it is yet unclear how they participate in the processing of information. Taking advantage of the ability of superior cervical ganglion neurons to form cholinergic autaptic contacts in the absence of glial cells[19,20], we investigated the neuronal reaction to synapse elimination caused by exposure to a C-terminal fragment of SPARC. The finding of a compensatory response based in the addition of new release sites coordinated with a potentiation of presynaptic calcium influx sheds light on how autaptic circuits maintain a constant synaptic output.

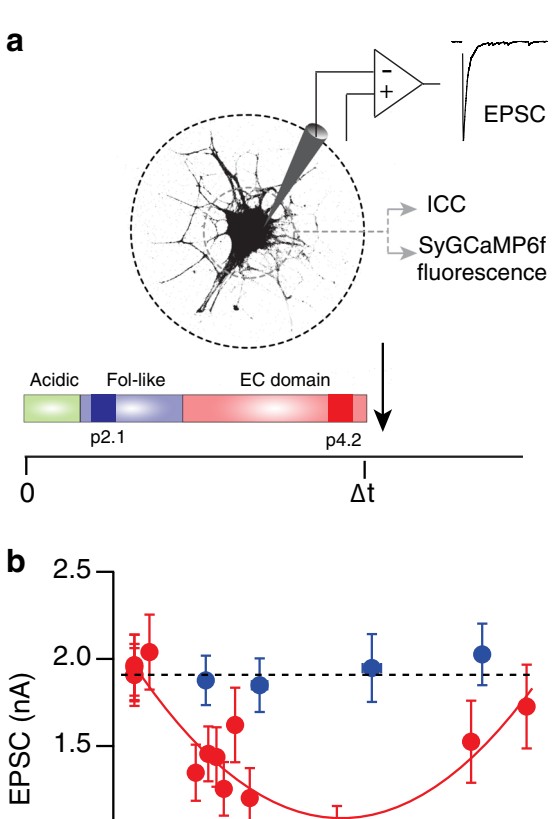

**Fig. 1 Exposure to p4.2 induces a biphasic response in synaptic strength.**
**a** Experimental setup. Evoked excitatory postsynaptic currents (EPSCs) were recorded from cholinergic single cell microcultures (SCMs). Electrophysiological recordings were associated to immunocytochemistry (ICC) for synaptic markers or to imaging of presynaptic calcium influx using synaptophysin:: GCaMP6f (SyGCaMP6f). SCMs were exposed for variable time intervals to p4.2, a 20 amino acid peptide derived from the EC-domain of SPARC, which triggers a cell-autonomous mechanism of synapse elimination. The 20 amino acid peptide p2.1, derived from the SPARC follistatin-like domain, was used as control. **b** A parabolic fit, $EPSC(t) = 3.49t^2 - 109.4t + 1960$, described changes in EPSC amplitude (pA) as a function of p4.2 exposure (t, hours). EPSCs decayed maximally after 15.4 h of exposure to 200 nM p4.2. Longer application times induced a compensatory response that returned EPSCs to control values within 30 h. Each bin indicates the average EPSC amplitude of ten individual neurons. Error bars show s.e.m. ($n = 280$ SCMs, 50 different cultures). Exposure to p2.1 ($n = 72$ SCMs, nine different cultures) did not induce any change in basal synaptic strength (dotted line). Each bin shows the average EPSC amplitude of 18 neurons. Error bars indicate s.e.m.

## Results

**Peptide p4.2 induces a biphasic change in synaptic strength.** Previous observations showed that exposure of cholinergic single cell microcultures (SCMs) for 2–5 h to 200 nM p4.2, a 20 amino acid peptide derived from the EC-domain of SPARC, activates a process of synapse elimination[13]. To find out if all autaptic contacts established by a single neuron could be disassembled or, if in contrast, there were synapses resistant to elimination, the incubation time with p4.2 was prolonged up to 30 h (Fig. 1a). Synapse loss was estimated by quantifying the decay in synaptic strength, measured as the amplitude of excitatory postsynaptic currents (EPSCs). Neither a complete suppression of synaptic transmission, nor a stabilization at a steady-state phase was observed. Synaptic strength reached a maximum decrease followed by a compensatory increase that was complete within ~30 h. Changes in EPSC amplitude were well described by a parabolic fit

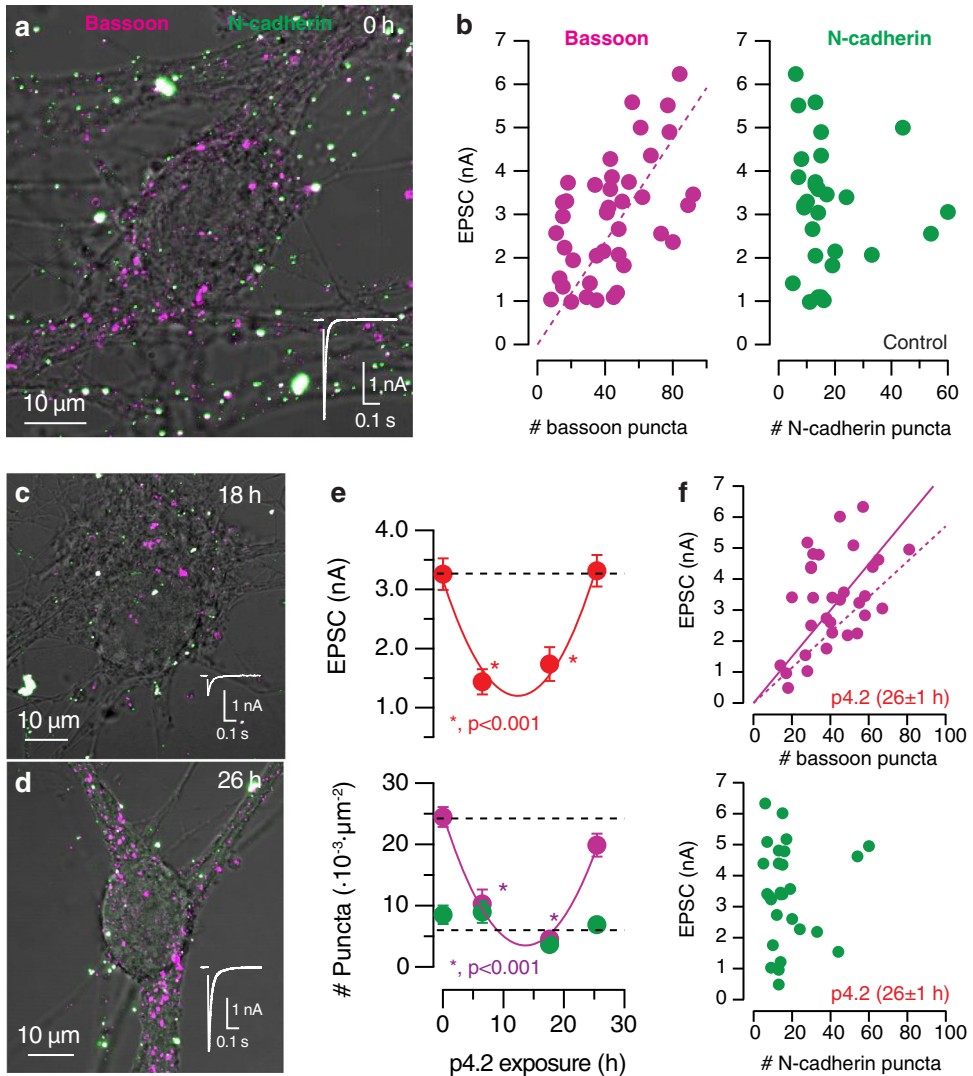

**Fig. 2 Rapid synapse formation compensates synaptic elimination caused by p4.2.** Correlative electrophysiology and immunocytochemistry experiments evaluated how changes in the density of synaptic puncta affected neurotransmission. **a** Image of the somatic and perisomatic region (soma + 20 μm radius) of a SCM stained for bassoon (magenta) and N-cadherin (green). Puncta found within the imaged area were evaluated as putative synapses contributing to the recorded excitatory postsynaptic current (EPSC, white trace). **b** Plots showing the relationship between EPSC amplitude and the number of bassoon or N-cadherin puncta detected in control SCMs ($n = 41$). Only bassoon puncta were linearly related to synaptic strength (59 pA·basson puncta$^{-1}$, Pearson correlation coefficient = 0.54). **c, d** Images of the somatic and perisomatic region (soma + 20 μm radius) of two SCMs exposed to p4.2 for 18 and 26 h, respectively. Both images display the associated EPSCs. **e** Relationship of EPSC amplitude as well as density of bassoon and N-cadherin puncta with p4.2 exposure time. Dots indicate mean ± s.e.m. Experimental groups: 0 h p4.2 (control), $n = 28$; 6 h p4.2, $n = 19$; 17.6 h p4.2, $n = 18$; 26 h p4.2, $n = 33$. Synaptic strength (expressed in pA) and the density of bassoon puncta were both related to p4.2 exposure time by two parabolic functions ($EPSC(t) = 13t^2 - 317t + 3162$, $Puncta(t) = 0.11t^2 - 3.14t + 25$). They showed minimum values at 12.4 and 15 h, respectively. Asterisks indicate significant differences relative to control values using one-way ANOVA followed by Bonferroni multiple comparisons test. **f** EPSC amplitude was linearly related (solid line) to the number of bassoon puncta after exposure to p4.2 for 26 h ($n = 33$, 72 pA·bassoon puncta$^{-1}$, Pearson correlation coefficient = 0.45). The dotted line indicates the relationship obtained for control SCMs (as in **b**), for comparison purposes. No relationship between synaptic strength and the number of N-cadherin puncta was found after exposure to p4.2 for 26 h ($n = 33$).

(Fig. 1b, $n = 280$). The alterations observed in neurotransmission were not related to changes in neuronal excitability, which was unaffected by exposure to p4.2 (Supplementary Fig. 1). Peptide p2.1, which also contains 20 amino acids but is derived from the SPARC follistatin-like domain, did not modify EPSC amplitude and was used as control (Fig. 1b, $n = 72$)[13].

The biphasic change in synaptic strength observed during exposure to 200 nM p4.2 suggested the activation of a homeostatic, compensatory mechanism of synapse elimination. To investigate this possibility, we carried out correlative electrophysiology and immunocytochemistry experiments. SCMs were

fixed and stained for bassoon and N-cadherin, a presynaptic active zone component and a synaptic adhesion protein, respectively (Fig. 2a). Although both markers showed a punctate staining, only 35% of bassoon labels co-localized with N-cadherin, suggesting the tagging of different structures. Bassoon labeling showed above 80% co-localization with the presynaptic markers synapsin-I and VAMP-2, suggesting a bona fide identification of synaptic terminals (Supplementary Fig. 2). The number of bassoon puncta located in the cell body and nearby dendritic tree (see "Methods" for details) was linearly related to synaptic strength, implying each identified contact contributed

with 59 pA to the recorded EPSC (Fig. 2b). Considering a quantal size of approximately 70 pA, the average probability of neurotransmitter release in a given autapse would be ~0.8, which is in agreement with previous observations[21]. Puncta identified by N-cadherin staining were, however, not linearly related to synaptic strength. This result could arise from the labeling of non-synaptic structures, or, could also reflect the dispensability of N-cadherin in the formation and stabilization of autapses present in SCMs, contrary to those formed by CNS neurons on dendritic spines[22].

The changes experimented in the density of bassoon puncta during exposure to 200 nM p4.2 paralleled variations in synaptic strength (Fig. 2c–e). Parabolic fits showed that the minimum EPSC amplitude and density of bassoon puncta were reached upon 12.4 and 15 h of p4.2 exposure, respectively. Changes in synaptic strength thus preceded the compensatory increase in the number of bassoon puncta and supported that synapse formation took place in a time window of ~2.5 h. New release sites were probably not the unique contributors to the homeostatic response because the original density of bassoon puncta was not completely recovered when EPSCs reached control values (Fig. 2e). The compensatory response likely involved a potentiation of synapses resistant to elimination. The slope of the linear relationship found between EPSC amplitude and the number of bassoon puncta of SCMs exposed to 200 nM p4.2 for more than 24 h (Fig. 2f) indicated that each synapse contributed with approximately 72 pA to neurotransmission, which represented approximately a 25% increase from control neurons. Synaptic strength found at the end of the homeostatic response continued being unrelated to the number of N-cadherin puncta. Altogether, correlative electrophysiology and immunocytochemistry experiments showed that autapses responded to elimination triggered by p4.2 through a coordinated potentiation of synaptic transmission and de novo synapse formation.

**Increased presynaptic calcium influx compensates synapse loss.** To evaluate the mechanisms accounting for the gain in synaptic function, we first investigated the size of the Readily Releasable Pool (RRP) of synaptic vesicles using high frequency stimulation[19,23,24]. The RRP is defined by those vesicles located at active zones that are ready to be released upon the arrival of an action potential. An increase in the RRP size, and thus, in the availability of synaptic vesicles could favor the enhancement of synaptic strength as reported in previous studies[25,26]. The RRP size experimented a biphasic change during exposure to p4.2, reaching the minimum value after 16 h (Fig. 3a). The compensatory response was, however, incomplete in the time interval explored, resembling results obtained for bassoon puncta (Fig. 2e). The study of RRP size was complemented by investigating short-term plasticity. The lower is the number of synaptic vesicles available for release, the higher is the depression assayed by paired pulse stimuli[1]. Depression evoked for a time interval of 50 ms was not modified during the phase of net synapse elimination but, it was enhanced during the compensatory response (Fig. 3b). Paired pulse depression increased by ~25% in SCMs treated for >24 h with p4.2, which could be explained by the participation of newly formed synapses. Immature contacts contain a low number of synaptic vesicles in their active zones[27] and tend to display short-term depression[28]. An alternative explanation for the increase in paired pulse depression could also be an increase in neurotransmitter release probability driven by the enhancement of presynaptic calcium influx.

To evaluate this possibility, we recorded EPSCs concomitantly to imaging of presynaptic calcium entry using SyGCaMP6f (Fig. 4a, Supplementary movie 1). The indicator accumulated in discrete puncta, which responded to the application of a train of five stimuli

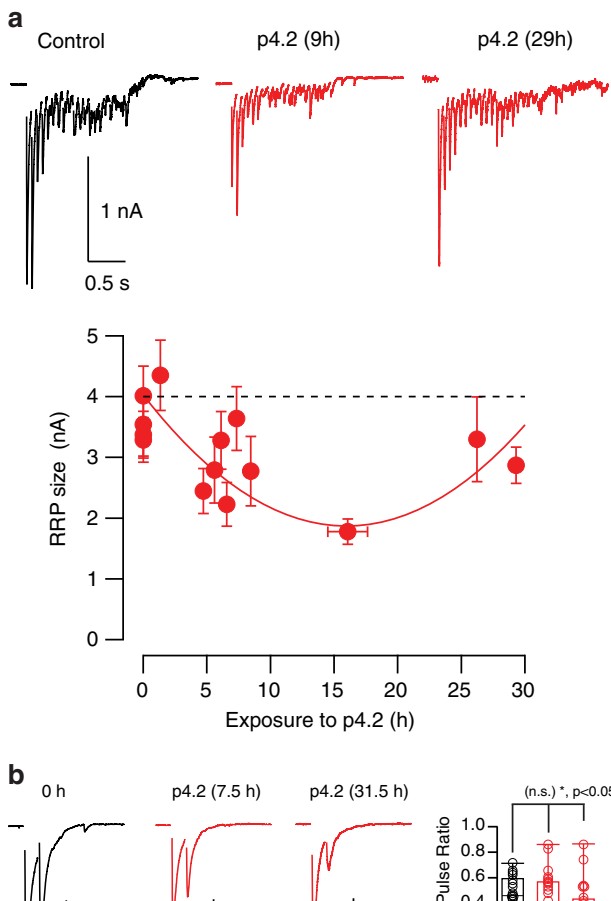

**Fig. 3 Changes in the size of the RRP and paired pulse plasticity during exposure to peptide p4.2. a** Trains of 20 stimuli delivered at 14 Hz were used to calculate the size of the RRP. Changes in the Ready Releasable Pool (RRP) size were biphasic and were described by a parabolic fit expressed in pA ($RRP(t) = 8.5t2 - 270t + 4028$). The minimum value was found after 16 h of exposure to peptide p4.2 ($n = 224$, each bin shows the mean ± s.e.m. of 16 different neurons). **b** Paired pulse plasticity was assayed by delivering two pulses at a time interval of 50 ms at the indicated times of p4.2 exposure. The characteristic depression observed in control neurons was increased at the end of the homeostatic response. Box plot shows the median (horizontal line), 25–75% quartiles (boxes), and ranges (whiskers) of the paired pulse ratio obtained in control neurons ($n = 18$), neurons exposed to p4.2 for 6.4 ± 1 h (mean ± s.e.m., $n = 26$) and neurons exposed to p4.2 for 27 ± 1 h (mean ± s.e.m., $n = 25$). Statistical differences were evaluated using one-way ANOVA followed by Bonferroni multiple comparisons test.

delivered at 20 Hz. The identity of putative synaptic contacts identified with SyGCaMP6f was confirmed after application of 1 μM TTX, which suppressed transient changes in fluorescence (Fig. 4b). The distribution of presynaptic calcium increases was well described by three gaussian functions showing mean ± SD $\Delta F/F_0$ changes of 0.022 ± 0.01, 0.100 ± 0.05 and 0.277 ± 0.02, respectively. The three populations were considered to represent terminals of low, medium, and high calcium influx, respectively (Fig. 4c). Since the classification of a given terminal could fluctuate as a result of changes in the open probability of calcium channels, we next evaluated the variability of SyGCaMP6f responses to single stimuli. It was not possible to resolve fluorescence increases in contacts that displayed low presynaptic calcium entry. Only

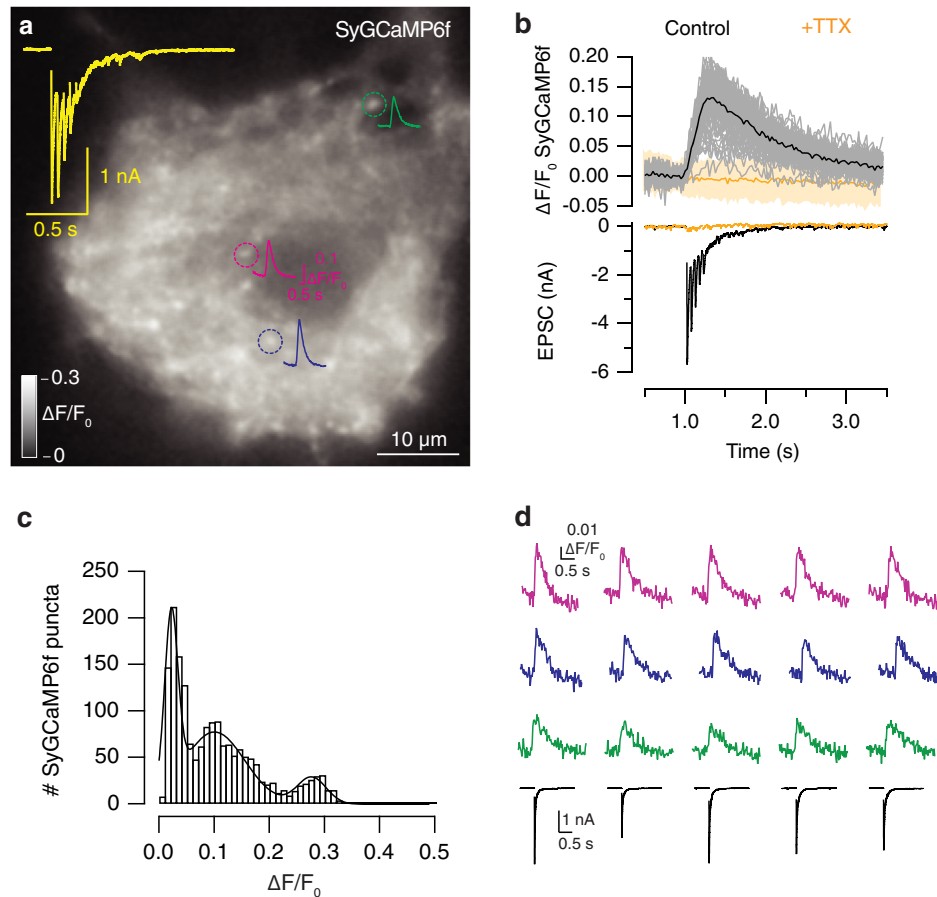

**Fig. 4 Characterization of presynaptic calcium influx in individual autaptic contacts using SyGCaMP6f. a** Increase in SyGCaMP6f fluorescence experimented by a SCM upon application of a train of five stimuli delivered at 20 Hz. The image is an average of 32 frames acquired at 40 Hz from the beginning of stimulation. Discrete, round ~1 μm spots that appeared as changes larger than three standard deviations of basal $\Delta F/F_0$ were considered as functional presynaptic terminals. Dotted circles show the responses of three different synapses. **b** ROIs displaying increases in SyGCaMP6f fluorescence during stimulation (individual synapses, gray traces; average, black trace) did not respond in the presence of 1 μM TTX (orange). The solid orange trace indicates the average response of identified synaptic contacts and shadowed area shows three times the SD. Notice that EPSCs were suppressed in the presence of TTX. **c** Distribution of peak $\Delta F/F_0$ changes detected in individual puncta in response to a train of five stimuli. Bars show data from control SCMs ($n = 1755$, 27 neurons). Fit to gaussian functions identified three population of responses with mean $\Delta F/F_0$ values of 0.02, 0.1, and 0.28, respectively. **d** Five consecutive responses of the presynaptic terminals identified in **a** to a single depolarization applied at 0.03 Hz. Notice that the increases in SyGCaMP6f fluorescence remained constant while the amplitude of the associated excitatory postsynaptic currents (EPSCs) varied.

synapses showing medium and high presynaptic calcium influx were analyzed. The amplitude of detected SyGCaMP6f transients was stable (CV = 8%, $n = 3$ SCMs), compared to the larger variability found in associated EPSCs (CV = 20%, $n = 3$ SCMs, Fig. 4d). This result indicates that: (i) a reliable opening of voltage gated calcium channels occurred independently of the intrinsic variations in neurotransmitter release probability, and (ii) global presynaptic calcium influx was an inherent property of identified contacts. Altogether, these results validated the presence of three different populations of synapses.

The characteristic biphasic response in synaptic strength to 200 nM p4.2 was also observed in SCMs expressing SyGCaMP6f (Fig. 5a) but, it was faster than in non-infected neurons (compare to Fig. 1b). The minimum EPSC amplitude was found at 9.4 h and the accomplishment of the compensatory response was observed upon ~18 h exposure. SCMs infected with lentiviruses were treated with maximal concentrations of CNTF (see "Methods" for details) to facilitate maximal synapse formation[29], which could explain the different kinetics of the homeostatic response. Treatment with p2.1 did not cause any alteration in synaptic strength and neither modify the probability of finding terminals with low, medium, or high calcium influx (Fig. 5b). The

amplitude of presynaptic calcium transients in response to a train of five stimuli also remained unaffected after 6 h of p4.2 exposure, however, it was modified during the compensatory response to synapse elimination. There was an overall increase in presynaptic calcium influx after 19 h of exposure to 200 nM p4.2 (Fig. 5c). The populations of contacts showing low, medium and high calcium responses were identified at mean ± SD $\Delta F/F_0$ changes of 0.054 ± 0.02, 0.177 ± 0.04, and 0.36 ± 0.04 (Fig. 5d), which suggested a broad potentiation of synapses resistant to p4.2. The increase in presynaptic calcium influx was also evident in responses evoked by single stimuli. For example, Fig. 5e illustrates a neuron exposed for 19 h to p4.2 that displayed a comparable synaptic strength to a control neuron but, showed larger presynaptic calcium transients. On average the amplitude of $\Delta F/F_0$ SyGCaMP6f changes evoked by single stimulation raised from 3.8 ± 0.4% ($n = 13$) to 5.6 ± 0.4 (mean ± s.e.m., $n = 15$, $p = 0.009$, unpaired $t$-test) by the end of the compensatory response, thus confirming a gain in presynaptic calcium entry.

**Synapse loss does not alter the density of nicotinic receptors**. To find out whether the compensatory increase of synaptic

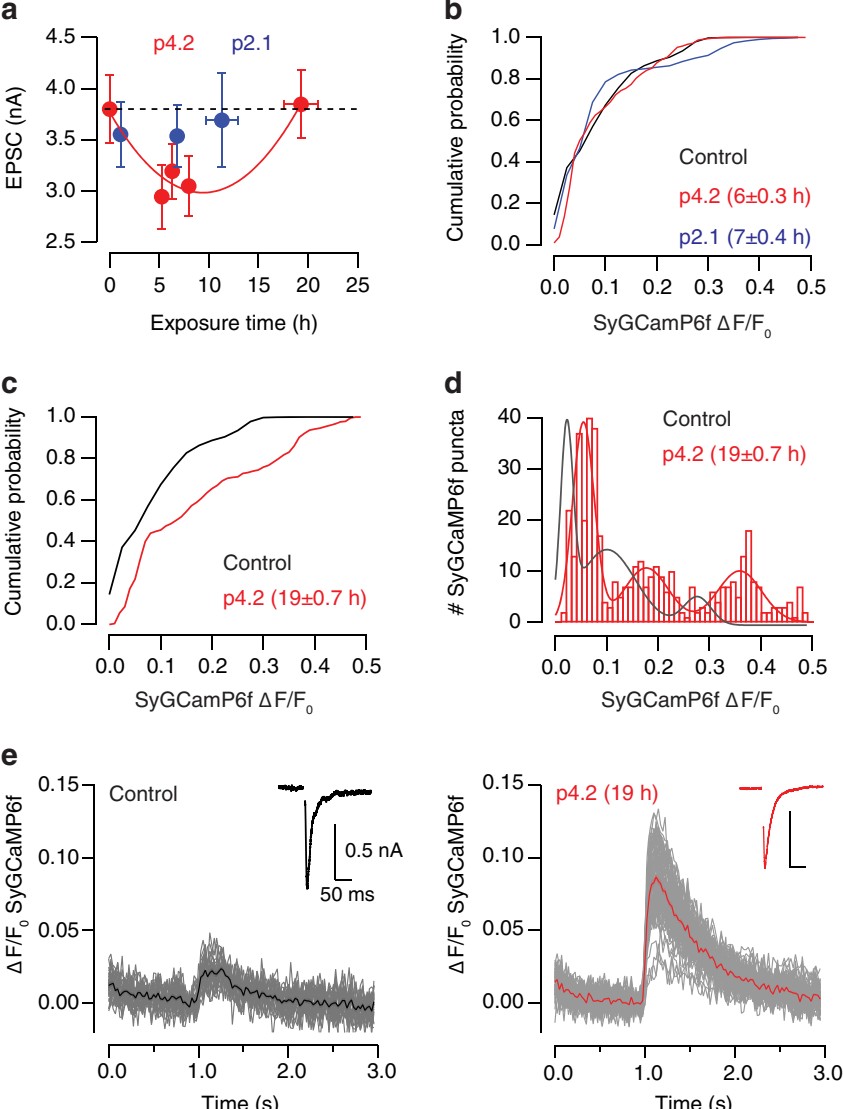

**Fig. 5 Exposure to p4.2 causes an increase in presynaptic calcium influx. a** Changes in EPSC amplitude during exposure of SCMs expressing SyGCaMP6f to 200 nM p4.2 or 200 nM p2.1. Changes in synaptic strength during treatment with peptide p4.2 followed a parabolic function showing a minimum value at 9.4 h ($EPSC(t) = 3.6t^2 − 112t + 1961$, expressed in pA). Bins indicate mean ± s.e.m. of 11 SCMs, ($n = 55$). Notice that exposure to p2.1 did not modify EPSC size (bins indicate mean ± s.e.m. of 8 SCMs, $n = 24$). **b, c** Cumulative probability of increases in SyGCaMP6f $\Delta F/F_0$ found in individual puncta upon application of a train of five stimuli delivered at 20 Hz. Data are illustrated for two time intervals of p4.2 exposure, which correspond to the minimum value of synaptic strength (**b** 6 ± 0.3 h, $n = 847$, 20 SCMs) and to the completion of the compensatory response (**c** 19 ± 0.7 h, $n = 521$, 10 SCMs), respectively. Time of exposure to p4.2 is indicated as mean ± s.e.m. Responses obtained in control conditions or in cells treated with p2.1 ($n = 647$) are displayed for comparison purposes. **d** Histogram showing individual peak $\Delta F/F_0$ changes obtained upon stimulation with a train of five stimuli delivered at 20 Hz ($n = 521$, 10 SCMs) in cells exposed to p4.2 for 19 ± 0.7 h (mean ± s.e.m.). They were described by three gaussian functions showing mean $\Delta F/F_0$ values of 0.05 and 0.18 and 0.36. The gray trace shows the scaled profile of the distribution of control responses for comparison purposes. **e** SyGCaMP6f responses of two different SCMs that showed comparable synaptic strength. Left, control neuron. The average response is shown in black. Right, presynaptic calcium increases recorded at the end of the compensatory response to synapse elimination. The average response is shown in red.

strength had also a postsynaptic component, we evaluated the contribution of nicotinic receptors to neurotransmission. Recordings obtained by electrical stimulation followed by the application of a local puff of 50 µM acetylcholine showed that the amplitude of EPSCs was comparable to the amplitude of chemically evoked nicotinic currents (Fig. 6a). The relationship between both types of postsynaptic currents was altered by p4.2 exposure. The characteristic decrease in synaptic strength was not followed by a reduction in the amplitude of nicotinic currents, which remained constant throughout the period investigated (Fig. 6b). Exposure to peptide p2.1 did not modify the response to

chemical or electrical stimulation (Fig. 6c). This result suggested that peptide p4.2 caused synapse elimination by driving the removal of the presynaptic element without affecting the density of postsynaptic cholinergic receptors.

Since local delivery of acetylcholine acts on both synaptic and extrasynaptic nicotinic receptors, we next investigated the role of synaptic receptors by analysing quantal size. The amplitude of miniature excitatory postsynaptic currents (mEPSCs) remained unaltered in SCMs exposed to p4.2 for different time intervals (Fig. 6d, e). A uniform average quantal size of ~70 pA, which was similar to previously reported values[19,30], demonstrated that the

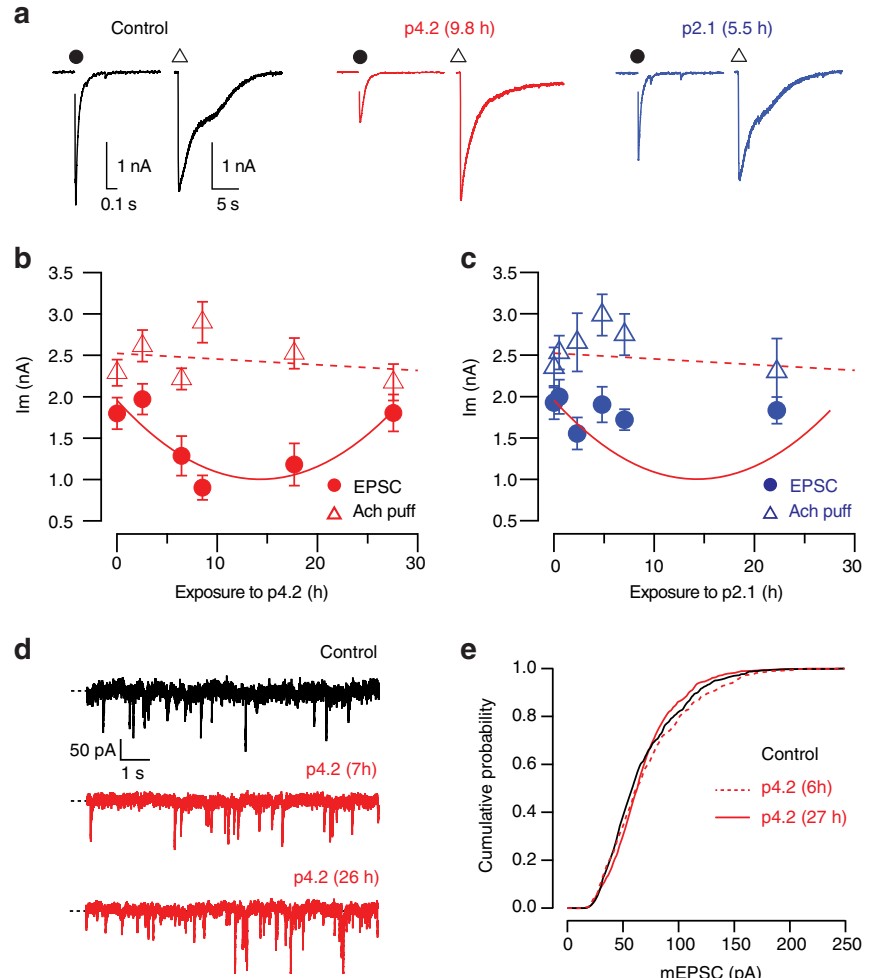

**Fig. 6 Separation of the pre and postsynaptic components of the compensatory response to p4.2 action. a** Recordings of evoked neurotransmission obtained by a sequential depolarization (dot) and a local puff of 50 µM acetylcholine (triangle). Time interval between electrical and chemical stimulation was 1 min. The examples illustrate the characteristic responses obtained in control SCMs or, upon exposure to p4.2 and p2.1. **b** Whilst the amplitude of EPSCs decayed during exposure to p4.2 following a parabolic fit (solid line), postsynaptic responses to acetylcholine remained unaltered (dotted line). Each bin indicates the mean ± s.e.m of ten different neurons ($n = 60$). **c** Neither EPSC amplitudes nor responses to an acetylcholine puff were modified by exposure to p2.1. Each bin shows the mean ± s.e.m of 8 different SCMs ($n = 48$). **d** Examples of miniature excitatory postsynaptic currents (mEPSCs) recorded in a control neuron and at the indicated times of exposure to 200 nM p4.2. **e** Cumulative probability of mEPSC amplitudes obtained in control neurons ($n = 574$, 12 SCMs) and in neurons exposed to p4.2 for an average time of 6 h ($n = 605$, 12 SCMs) or 27 h ($n = 607$, 10 SCMs).

number of synaptic nicotinic receptors was essentially unaffected by p4.2 action. The absence of an obvious postsynaptic alteration makes unlikely that cholinergic receptors were the trigger of the homeostatic response, as it occurs in other neuronal types[31]. Moreover, the maintenance of a constant density of cholinergic receptors in the plasma membrane could facilitate the assembly of new contacts during the compensatory response to synapse loss.

**Peptide p4.2 remodels the neuronal F-actin cytoskeleton.** Peptide p4.2 displays counteradhesive properties on cultured endothelial cells due to its ability to disrupt focal adhesions[32,33] by presumably recapitulating the ability of SPARC to redistribute F-actin[34]. To investigate if p4.2 was capable of altering the organization of F-actin in other cell types, we exposed CHO cells to 200 nM p4.2 for 16 h (Fig. 7a). As a result, there was an enrichment of filamentous actin in the periphery of cells (Fig. 7b, c), which was not related to depolymerization. Exposure to latrunculin-A increased soluble actin monomers in the cell cytoplasm, however, peptide p4.2 failed to do so and confirmed its ability to reorganize the F-actin cytoskeleton without

disrupting it (Fig. 7d). Comparable results were obtained using HEK-293 cells and confirmed the capacity of peptide p4.2 to induce a reorganization of the F-actin cytoskeleton.

We next assessed if peptide p4.2 displayed a similar action on the actin network found in neurons. STED microscopy revealed the characteristic periodic F-actin skeleton of neuronal processes (Fig. 8a–c). Actin ring-like structures showed a periodicity of 195 nm, which is comparable to values described for cultured hippocampal neurons[35–37]. Exposure of SCMs to 200 nM peptide p4.2 for 15 or 27 h disrupted actin organization: F-actin appeared distributed in patches rather than in regularly spaced bands (Fig. 8d, e). Although there was no obvious organization of F-actin, some neuritic regions showed periodic structures located at a distance between 250 and 300 nm.

To estimate the time course of changes in the organization of the F-actin cytoskeleton present along the axon and dendrites, we took advantage of the ability of latrunculin-A to increase neurotransmitter release probability[38]. The analysis of responses to paired pulse stimuli showed that disruption of presynaptic F-actin by local application of 20 µM latrunculin-A[39] enhanced

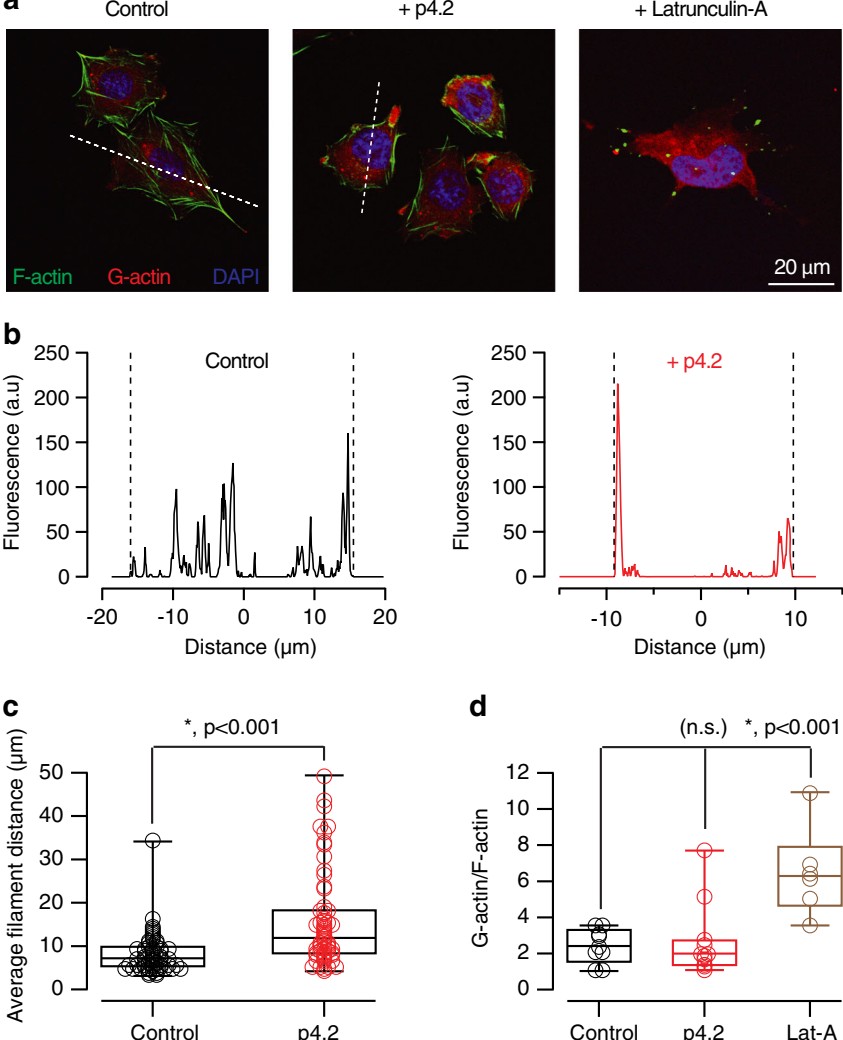

**Fig. 7 Exposure to p4.2 causes a reorganization of the F-actin cytoskeleton. a** Images of CHO cells labeled with phalloidin-Alexa Fluor 488 and deoxyribonuclease I-Alexa Fluor 594 to show the distribution of F-actin and G-actin, respectively. Cells were treated for 16 h with 200 nM p4.2 or 1 μM latrunculin-A. **b** A line profile (dotted line in **a**) was drawn through the longest cell axis (see "Methods" for details) to evaluate the distribution of F-actin. Fluorescence peaks indicate the distribution of actin filaments. The cell center is set to a distance of 0 μm and cell borders are indicated by dotted lines. The examples show that F-actin concentrated in the periphery as a result of p4.2 treatment. **c** Box plot showing the median (horizontal line), 25–75% quartiles (boxes), and ranges (whiskers) of distance among F-actin filaments in control cells ($n = 58$) and cells exposed to p4.2 ($n = 60$). Statistical differences were evaluated using unpaired two-tailed Student's $t$-test. **d** The relationship between G-actin and F-actin was studied on a single cell basis by integrating the fluorescence of deoxyribonuclease I-Alexa Fluor 594 and phalloidin-Alexa Fluor 488. Box plot shows the median (horizontal line), 25–75% quartiles (boxes), and ranges (whiskers) of G-actin/F-actin in control cells ($n = 9$), cells exposed to p4.2 ($n = 11$) and cells treated with latrunculin-A ($n = 6$). Evaluation of statistical differences using one-way ANOVA followed by Bonferroni multiple comparisons test showed that exposure to p4.2 did not modify the relationship between G-actin and F-actin. Latrunculin-A was used as a control for F-actin depolymerization.

synaptic strength and, consequently, increased short-term depression by ~40% (Fig. 9a). Although basal paired pulse plasticity assayed in a 100 ms interval was not significantly decreased by exposure to p4.2, depression evoked by latrunculin-A was indeed enhanced (Fig. 9b and c). In 3 out of 7 cells treated for $19 \pm 1$ h (mean ± s.e.m.) with p4.2, the second EPSC was almost absent, due to a maximum increase of release probability. Taking into account that short-term depression is essentially determined presynaptically[1], we used the potentiation of latrunculin-A effect as a readout of F-actin remodeling induced by p4.2. The decrease in paired pulse ratio as a result of F-actin depolymerization decayed exponentially as a function of p4.2 exposure (Fig. 9d), suggesting that the peptide instructed a reorganization of presynaptic actin with a time constant of 5.5 hours. A complete rearrangement of the F-actin cytoskeleton

could be expected after three time constants, which coincides with the maximum synapse loss found at ~16 h (Fig. 2e).

**F-actin stabilization prevents synapse elimination.** To investigate whether reorganization of the F-actin cytoskeleton was driving synapse elimination and the consequent homeostatic response, the action of p4.2 was impeded by simultaneous exposure to jasplakinolide, an F-actin stabilizing drug[13,40]. Quantification of the number of SyGCaMP6f puncta responding to a train of five stimuli confirmed the ability of jasplakinolide to prevent synapse loss. Control SCMs contained $(2.23 \pm 0.30) \times 10^{-3}$ puncta·μm$^{-2}$ ($n = 27$, mean ± s.e.m.), while neurons exposed for 6 h to p4.2 presented a lower density of $(1.26 \pm 0.17) \times 10^{-3}$ puncta·μm$^{-2}$ ($n = 20$; Fig. 10a, $p < 0.05$). As for correlative electrophysiology and immunocytochemistry

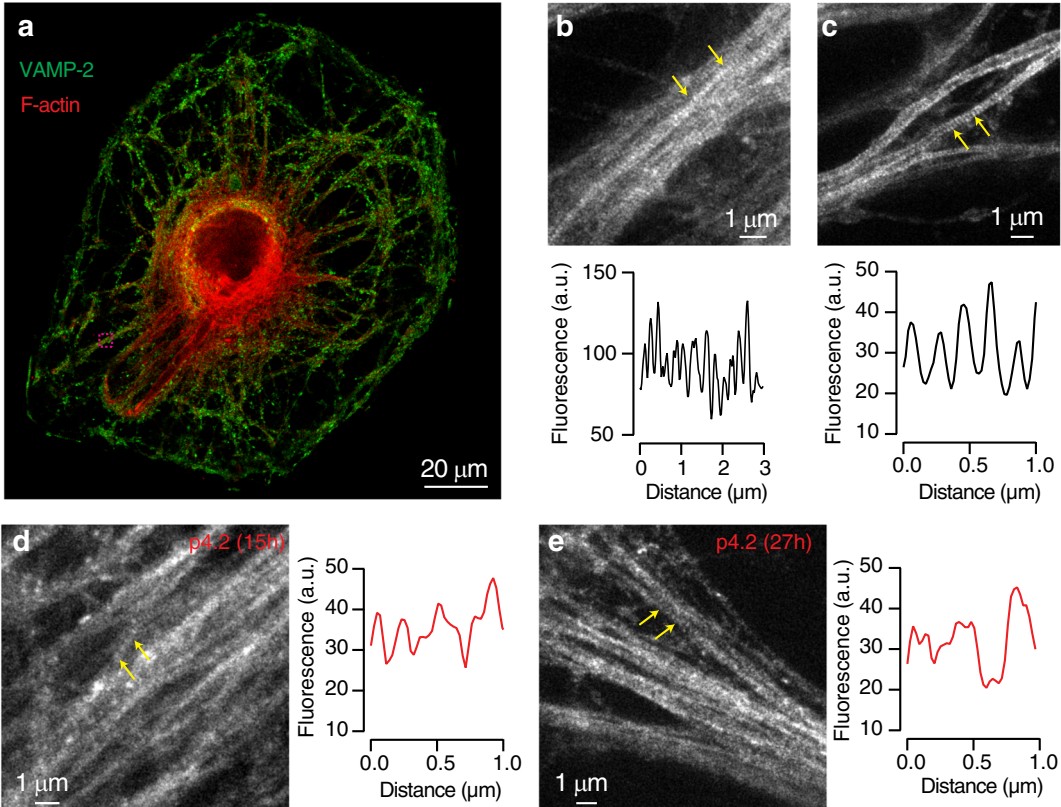

**Fig. 8 Peptide p4.2 disrupts the periodic actin skeleton present in neurites. a** Image of a SCM showing the distribution of VAMP-2 and F-actin stained with phalloidin-Atto 647 N. The boxed region was selected for visualization with STED microscopy (shown in **b**). **b, c** Up, periodic actin structures are present along neurites of control SCMs. Images were obtained in two different neurons. Down, plot of line profiles drawn between arrows. The presence of actin-ring like structures is illustrated by peaks of fluorescence, which are regularly spaced at a distance of ~190 nm. **d, e** The overall appearance of the periodic actin skeleton is altered by exposure to 200 nM peptide p4.2 for 15 h (**d**) or 27 h (**e**). Most neurites do not show obvious actin-ring like structures, however, plot profiles in selected neuronal processes (arrows) display regularly spaced peaks at 250–300 nm distance.

experiments (Fig. 2), the density of SyGCaMP6f puncta paralleled changes in synaptic strength, increasing to $(1.86 \pm 0.30) \times 10^{-3}$ puncta·μm$^{-2}$ ($n = 10$) after 19 h of p4.2 treatment. The relationship found between synaptic strength and the density of SyGCaMP6f puncta was maintained when p4.2 was applied concomitantly to jasplakinolide. Neither EPSC amplitude, nor the density of $(2.12 \pm 0.28) \times 10^{-3}$ puncta μm$^{-2}$ ($n = 16$) were different from control neurons. The distribution of changes in SyGCaMP6f fluorescence was also comparable to controls (Fig. 10b). These data suggest that the stabilization of the actin cytoskeleton precluded synaptic elimination induced by p4.2 and the consequent potentiation of presynaptic calcium influx.

## Discussion

In the current work, we characterize the neuronal response activated to counterbalance the cell-autonomous elimination of synaptic contacts triggered by p4.2, a peptide derived from the C-terminal region of SPARC[13]. Neurons grown in an autaptic circuit are capable of maintaining baseline neurotransmission without modifying their number of postsynaptic receptors. The homeostatic response exclusively invokes presynaptic mechanisms that combine the potentiation of presynaptic terminal function with the assembly of new synaptic contacts. Presynaptic calcium influx and RRP size increase, which are both hallmarks of presynaptic homeostatic plasticity[26,31]; moreover, the rapid formation of a finite number of synapses demonstrates the ability of neurons to carry out synapse assembly as part of the compensatory response. These evidences together show that neurons

establishing an autaptic circuit set a baseline level of neurotransmission by sensing and finely adjusting the number and properties of neurotransmitter release sites.

The molecular mechanisms responsible for presynaptic homeostatic plasticity have been previously investigated using the alteration of postsynaptic receptors as a trigger (reviewed in the ref.[2]). In the current work, compensatory presynaptic mechanisms are not activated by the impairment of postsynaptic terminal function but, by the retraction of the presynaptic element in an autaptic circuit[13]. Yet, part of the homeostatic response can be considered classical because it involves the potentiation of presynaptic calcium influx (see for example ref.[41]), the participation of rapid synapse formation must also be accounted for. What signal could act as a trigger for the gain of presynaptic function as well as for the addition of new neurotransmitter release sites? The participation of trans-synaptic communication pathways is unlikely because the compensatory response is not activated by an alteration of the density of postsynaptic receptors[42]. An alternative must come from other signaling pathways. The modification of the periodic actin skeleton found in neuronal processes might be a key to de-stabilize synaptic contacts and trigger the formation of multivesicular bodies and other endocytic structures containing presynaptic elements[13]. Retrograde transport of formed lysosomal degradation compartments could be important for determining neuron survival[43,44] and in the light of our observations, could also be used for reporting synapse loss.

A quantitative estimate of the intensification of synaptogenesis occurring during the homeostatic response can be drawn

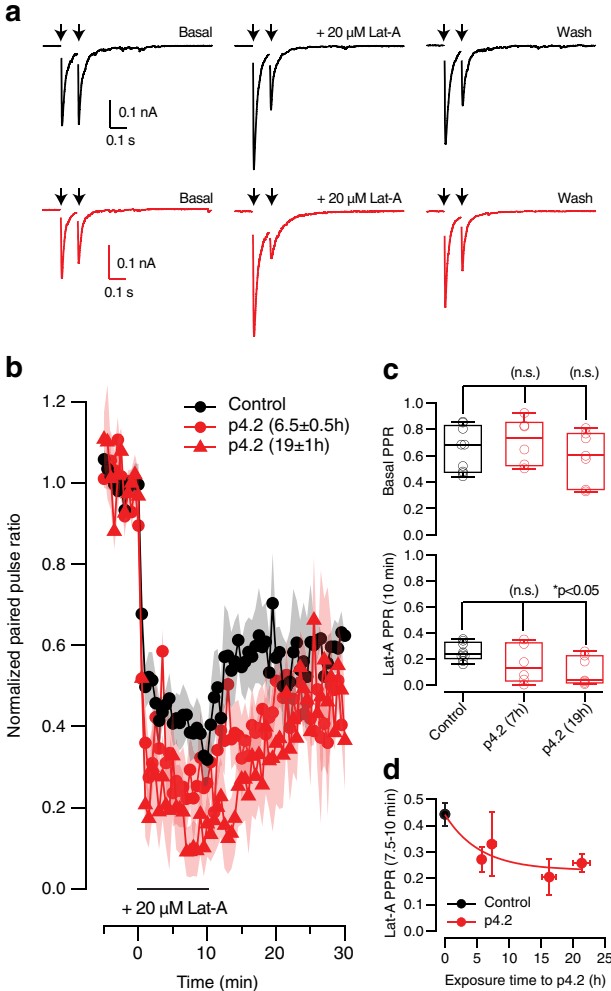

**Fig. 9 Time course of presynaptic F-actin cytoskeleton reorganization induced by peptide p4.2. a** Changes in short-term plasticity caused by latrunculin-A were used to evaluate the timescale of F-actin reorganization induced by p4.2. Application of 20 µM latrunculin-A enhanced paired-pulse depression measured in a 100 ms interval by increasing the amplitude of the first EPSC. A partial recovery of depression was observed after washing out latrunculin-A. Black traces show paired-pulse plasticity recorded in a control SCM. Red traces illustrate the effect of 20 µM latrunculin-A in a cell exposed to 200 nM p4.2 for 19 h. **b** Normalized paired-pulse ratio as a function of experiment time. Notice that depression was immediately enhanced upon application of latrunculin-A. **c** Paired pulse ratio (PPR) recorded in basal conditions (up) and 10 min after application of 20 µM latrunculin-A (down) in the indicated experimental conditions. Box plots show the median (horizontal line), 25–75% quartiles (boxes), and ranges (whiskers) of PPR values obtained in control conditions ($n = 9$) and 7 h ($n = 6$) or 19 h ($n = 7$) after application of 200 nM p4.2. **d** Decay in PPR caused by latrunculin-A as a function of exposure time to peptide p4.2. Each bin shows the mean ± s.e.m. of three different neurons ($n = 12$). Individual PPR values correspond to the average decrease observed between 7.5 and 10 min of latrunculin-A exposure. PPR obtained in control SCMs is indicated in black (mean ± s.e.m, $n = 9$). Bins are fitted to a single exponential function, providing a time constant of 5.5 h.

considering that a prototypical SCM contains ~200 synapses. Upon exposure to p4.2, the neuron would react to synapse elimination by forming ~75 new contacts in 10 h. If the assembly time of cholinergic autaptic synapses is found between ~1[45] and 2.5 h (see Fig. 2e), it would mean SCMs establish 3–8 new contacts hour$^{-1}$ in response to synapse elimination. Considering a

likelihood of finding the formation of a contact in cultured neurons of 0.4–1%[46], basal synapse formation could increase by almost an order of magnitude during the homeostatic response. In other terms, a neuron growing in a SCM takes a week to establish all its contacts[21] but, in front of a significant synapse loss, the neuron can recover one third of its synapses just in half a day.

By the end of the homeostatic response mature and recently assembled autapses coexist. Newly formed contacts likely contributed to the observed increase in RRP size, which could also be accomplished by the activation of certain regulatory mechanisms operating at the level of active zones[26]. The small RRP size of immature contacts[21,27] explains why synaptic strength recovers completely during p4.2 exposure, whilst RRP does not. Fast synapse formation allows the re-establishment of neurotransmitter release sites, which can efficiently support synaptic transmission but with a limited availability of synaptic vesicles. Short-term plasticity is consequently affected, shifted towards a depressing phenotype.

Assuming that the homeostatic response is orchestrated from the cell body, the increase in presynaptic calcium entry is probably mediated by the insertion of voltage gated calcium channel copies at the synapse. However, it remains an open question how is it possible to achieve the functional assembly of synaptic contacts upon disruption of neuritic ring-like F-actin structures. Changes in the molecular composition of voltage-gated calcium channels as well as, of certain accessory subunits or adhesion proteins present in the presynaptic terminal, could be important for establishing functional synapses in an altered neuronal cytoskeleton. Any molecular modification was, however, not determinant for setting functional differences among synaptic contacts. By the end of the compensatory phase, all synapses experimented a comparable potentiation of presynaptic calcium entry. This widespread gain of function could be related to the use of cultured neurons[4] but could also be reflecting a specific property of autaptic circuits.

Autaptic contacts are commonly established by excitatory and inhibitory neurons, such as pyramidal neurons of the neocortex, cerebellar interneurons or spiny neurons of the striatum[16–18,47,48]. Although the specific roles of autapses have been a matter of debate[49], there are growing evidences that support their implication in relevant network functions. For example, excitatory autapses are important to maintain persistent activation of B31/B32 neurons in Aplysia[50], or, to enhance bursting in layer 5-pyramidal neurons[51]. The coordinated potentiation of presynaptic terminal function with de novo synapse formation used for setting a correct level of neurotransmission might unmask a biologically relevant role of autapses. Excitatory neurons could use autaptic contacts to sense their synaptic output and take advantage of the mechanism here described to increase gain and, consequently enhance excitability. Future experiments should verify if excitatory autapses are indeed keepers of neuronal firing.

## Methods

**Molecular biology**. The synaptophysin::GCaMP6f (SyGCaMP6f) construct was provided by Dr. Leon Lagnado as an improved version of syGCaMP2[52,53]. The coding sequence of SyGCaMP6f was cloned into pWPXL (Addgene plasmid # 12257; http://n2t.net/addgene:12257; RRID:Addgene_12257) by replacing EGFP using BamHI and NdeI. For lentivirus production, HEK 293T cells were transfected by the calcium phosphate technique following methods described by Didier Trono (http://tronolab.epfl.ch/lentivectors) with pMD2G (Addgene plasmid # 12259; http://n2t.net/addgene:12259; RRID:Addgene_12259), pCMVR8.74 (Addgene plasmid # 22036; http://n2t.net/addgene:22036; RRID:Addgene_22036), and pWPXL. Two days later, culture medium containing lentiviral particles was collected in three rounds at 8 h intervals, kept at 4 °C, and centrifuged at 500×g. Supernatants were distributed in aliquots and stored at −80 °C.

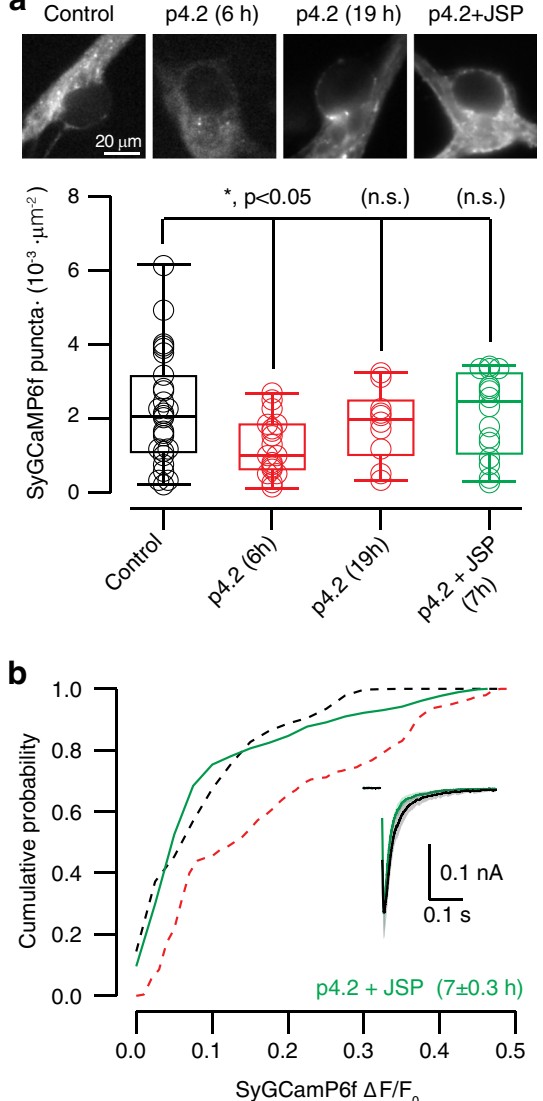

**Fig. 10 Stabilization of the presynaptic F-actin cytoskeleton prevents the action of p4.2. a** The transient decay of SyGCaMP6f puncta caused by exposure to 200 nM p4.2 was not observed when the peptide was incubated in the presence of 1 µM jasplakinolide. Examples of difference images obtained during stimulation with five stimuli delivered at 20 Hz in the indicated conditions. Notice that the density of putative synaptic contacts is transiently reduced during p4.2 exposure but remains unaltered if neurons are treated concomitantly with jasplakinolide. Box plots show the median (horizontal line), 25–75% quartiles (boxes), and ranges (whiskers) of the density of SyGCaMP6f puncta in control conditions ($n = 27$), after 6 h of exposure to p4.2 ($n = 20$), after 19 h of exposure to p4.2 ($n = 10$) and after 7 h incubation of p4.2 and jasplakinolide ($n = 16$). Statistical differences were evaluated using one-way ANOVA followed by Bonferroni multiple comparisons test. Only exposure to p4.2 for 6 h induced a significant decrease of SyGCaMP6f puncta. **b** Cumulative probability of SyGCaMP6f $\Delta F/F_0$ increases observed after application of a train of five stimuli delivered at 20 Hz in neurons exposed to 200 nM p4.2 and 1 µM jasplakinolide for 7 ± 0.3 h (mean ± s.e.m., $n = 1030$, 16 SCMs). Dotted lines show the distributions obtained in control conditions (black) and after exposure to peptide p4.2 for 19 h (as shown in Fig. 5c) for comparison purposes. The inset displays the profile of the average EPSC (solid line ± shadowed area indicate mean ± s.e.m.) obtained in control SCMs (black, $n = 12$) and in neurons treated with 200 nM peptide p4.2 and 1 µM jasplakinolide (green, $n = 11$).

**Cell culture**. Experimental procedures were approved by the Department of Environment from the Generalitat de Catalunya. SCMs from superior cervical ganglion neurons were prepared following previously described methods using postnatal day 0 (P0) to P2 Sprague Dawley rats[19,30]. Briefly, medium containing all dissociated ganglionic cells was placed in a 100-mm-diameter culture dish for 60 min at 37 °C. At the end of this preplating period, ≥95% of non-neuronal cells were found to be adhered to the dish, but most neurons remained in suspension. Medium was then collected, and neurons seeded at 2500 cells mL$^{-1}$ on 15 mm coverslips containing 10–20 collagen microdrops of 100–400 µm diameter. Culture medium was DMEM/F12 [1:1] containing 2.5% fetal bovine serum, 2.5% rat serum (prepared in the animal care facility of the Campus of Bellvitge, University of Barcelona), 5 nM NGF, 1–2 nM CNTF (Alomone Labs, Jerusalem, Israel), and 25 U/ml penicillin/streptomycin at 37 °C and 8% $CO_2$. To genetically modify SCMs, lentiviral infection was applied at 13 days in vitro (D.I.V.) during a 1:2 overnight incubation of the appropriate viral stock. Fluorescence was evident at 4–6 days after infection. SCMs were exposed for time intervals ranging from 0 to 30 h to p2.1 (CQNHHCKHGKVCELDESNTP) or p4.2 (TCDLDNDKYIALEEWAGCFG). Peptides p2.1 and p4.2 were synthesized by GL Biochem (Shanghai, China). Stocks were prepared at 2 mM using MQ water and aliquots were stored at −80 °C. Peptide solutions were thawed and diluted in culture medium.

CHO cells were grown on coverslips coated with the collagen used for establishing microcultures. Cells from passages 8–12 were seeded at a density of 20,000 cells mL$^{-1}$. Culture medium was DMEM/F12 [1:1] containing 10% fetal bovine serum and 10 U/ml penicillin/streptomycin. CHO cells were maintained at 37 °C and 5% $CO_2$.

**Immunocytochemistry and STED microscopy**. For immunocytochemistry, coverslips containing SCMs were fixed in 4% PFA prepared in phosphate buffer 0.1 M. They were incubated overnight at 4 °C with a monoclonal anti-bassoon primary antibody (1:1000, Enzo Life Sciences, ADI-VAM-PS003) combined with one of the following polyclonal antibodies: anti-N-cadherin (10 µg/ml, R&D Systems, AF6426), anti-VAMP-2 (1:500, Synaptic Systems, 104202), or, anti-Synapsin-I (1:1000, Millipore, AB1543P). Appropriate secondary antibodies labeled with Alexa Fluor 488 and Alexa Fluor 555 were used for fluorescent staining. Cells were visualized in a Zeiss LSM 880 confocal microscope (Carl Zeiss AG, Oberkochen, Germany). Optical sections were acquired using a 63× oil immersion objective PL-APO (1.4 N.A.).

For visualization by STED microscopy, SCMs were immersed for 15 min in a 4% PFA solution prepared in phosphate buffer 0.1 M. Overnight incubation at 4 °C with a primary antibody anti-VAMP-2 (1:1000, Synaptic Systems 104211) was followed by application of an anti-mouse secondary antibody labeled with Alexa Fluor 488. SCMs were subsequentially incubated with 0.165 µM phalloidin-Atto 647N (Sigma-Aldrich, 65906) for 2.5 h at room temperature, followed by an overnight period at 4 °C.

Single plane STED images were acquired on a Leica TCS SP8 STED 3× (Leica Microsystems, Mannheim, Germany) on a DMI8 stand using a 100×/1.4NA HCS2 PL APO objective. A pulsed supercontinuum light source set at 644 nm was used for excitation and a pulsed depletion laser at 775 nm was added with no pulse delay at 20% intensity. Detection was performed with a hybrid detector (HyD) between 652 and 750 nm and gating between 0, 3, and 6 ns. Scanner speed was set to 600 Hz and images were taken with 8× frame accumulation and 2× frame average. Pixel size was set according to the depletion power to 23 nm.

**Electrophysiological recordings**. All experiments were performed in the whole-cell configuration of patch-clamp mode using neurons microcultured for 18–22 D.I.V. Typical resistances of pipettes used for recordings were 3–5 MΩ when filled with internal solution composed of the following (in mM): 130 K-gluconate, 4 $MgCl_2$, 1 EGTA, 10 HEPES, 3 $Na_2ATP$, 1 NaGTP, pH 7.2, 290 mOsm/kg. External solution contained (in mM): 130 NaCl, 5 KCl, 2 $MgCl_2$, 10 HEPES-hemisodium salt, and 10 glucose, pH 7.4. The final 2 mM $CaCl_2$ concentration was always achieved by dilution from a 1 M stock. All salts were from Sigma-Aldrich (St. Louis, MO). Before the addition of glucose and $CaCl_2$, the osmolality of the external solution was adjusted to 290 mOsm/kg. Experiments were performed at room temperature (23 °C).

Recordings were made using an Axopatch-1D patch-clamp amplifier (Molecular Devices, San Jose, CA) under the control of an ITC-18 board (Instrutech Corp, Port Washington, NY) driven by WCP software (Dr. John Dempster, University of Strathclyde, UK) or mafPC (courtesy of M. A. Xu-Friedman, University at Buffalo, NY). Neurons were clamped at −60 mV and stimulated by a 1–2 ms depolarization step that drove membrane potential to 0 mV. The presence of functional autaptic synapses was identified by the generation of excitatory postsynaptic currents (EPSCs). Further details of neurotransmission in SCMs are described elsewhere[13,19,21,30]. To test the contribution of postsynaptic nicotinic receptors, a local puff of 50 µM acetylcholine was locally applied using a fused-silica capillary (Microfil, #MF-28G, World Precision Instruments, Sarasota, FL). Timing of application was set to 30 ms using a TTL pulse.

**Simultaneous electrophysiological recording and imaging**. To associate presynaptic calcium influx to neurotransmission, SCMs expressing SyGCaMP6f were

imaged and simultaneously recorded electrophysiologically (Fig. 1a), see also ref. [54]. Coverslips were mounted on an RC-25 imaging chamber (Warner Instruments, Hamden, CT) and observed in an inverted Olympus IX-50 microscope. Cells were illuminated with blue light, using an ET480/20x excitation filter. Fluorescence was acquired using a q505LP dichroic and a HQ535/50 nm emission filter (Chroma Technology Corp., VT). Images were collected through a 60× UPlanFLN objective (1.25 N.A, Olympus, Tokyo, Japan) and visualized on an ImageEM camera controlled by HCImage (Hamamatsu). Images from a $256 \times 256$ pixels box located on the perisomatic region were acquired at 40 Hz. TTL pulses generated by mafPC controlled the exposure time of the camera, as well as the timing of light illumination, which was adjusted via a shutter (Uniblitz, NY), to minimize photobleaching.

**Analysis of electrophysiological and imaging data**. Electrophysiological recordings were analyzed using custom-made macros written in Igor Pro software (Wavemetrics, Lake Oswego, OR) versions 6.0-8.0. Image analysis was carried out combining Image J and macros written in Igor Pro software. SyGCaMP6f fluorescence (F) was obtained after subtracting the background fluorescence found in regions of interest drawn outside of the microculture. Calcium increases were reported as relative changes of SyGCaMP6f fluorescence, measured as $(F - F_0)/F_0$. Putative synapses were identified on difference images that illustrated the location of stimulation dependent changes in fluorescence (see examples in Fig. 10a). $1.3 \times 1.3$ μm ROIs were drawn on the center of mass of identified puncta, which were typically round, ~1 μm diameter structures. A given punctum was considered a synapse when exhibited a $\Delta F/F_0$ increase above three standard deviations of baseline values recorded before stimulation. This criterion avoided the contribution of synapses found out of focus. Block of depolarization with 1 μM TTX ($n = 4$) demonstrated that changes in SyGCaMP6f fluorescence were exclusively related to neurotransmission (see the example in Fig. 4b).

**Correlative electrophysiology and immunocytochemistry**. SCMs displaying a representative morphology were recorded, micrographed, fixed, and stained by immunocytochemistry (Fig. 1a) for the expression of the presynaptic marker bassoon (Enzo Life Sciences, ADI-VAM-PS003) and the synaptic adhesion protein N-cadherin (R&D Systems, AF6426). Appropriate secondary antibodies labeled with Alexa Fluor 488 and Alexa Fluor 555 were used to visualize N-cadherin and bassoon labeling, respectively. Images of recorded SCMs were used to identify stained microcultures in a Zeiss LSM 880 confocal microscope. Confocal sections were acquired using a 63× oil immersion objective PL-APO (1.4 N.A.). The procedure for obtaining an estimate of synapses contributing to recorded EPSCs was based in the quantification of the density of bassoon and N-cadherin puncta located in the somatic region and nearby dendritic tree. Specifically, only somatic and dendritic puncta found ≤20 μm away from the edge of the soma were considered. Synaptic contacts more distally located were not considered because the cable-filtering properties of dendrites in cultured neurons attenuate their contribution to recorded EPSCs[55]. A punctum was considered as a putative synapse if it spanned from three to five consecutive confocal Z-sections (0.33 μm optical thickness), had a diameter ranging from 0.2 to 0.5 μm and was stained with bassoon or N-cadherin. Density was calculated by dividing the number of bassoon and N-cadherin puncta by the analyzed area.

**Quantification of F-actin and G-actin stainings**. Simultaneous labeling of G-actin and F-actin was carried out following previously described methods[56,57]. Briefly, CHO cells were fixed in PFA 4% and stained with phalloidin-Alexa Fluor 488 (A12379, Thermofisher, Waltham, MA) and deoxyribonuclease I-Alexa Fluor 594 (D12372, Invitrogen, Carlsbad, CA). Cells were visualized under a Leica TCS-SL confocal microscope using a 63× immersion oil objective PL-APO (1.4 N.A). Images were analyzed in Image J.

The distance among F-actin filaments was calculated using a line profile drawn along the longest cell axis estimated by Image J (Feret's diameter). Only the basal region of the cell (~1.5 μm optical thickness) was considered for analysis. To estimate the quantity of G-actin and F-actin in a single cell, the fluorescence of all pixels found in the maximum intensity projection were summed (RawIntDen command in Image J) after background subtraction. The relationship between G-actin and F-actin was calculated by dividing values obtained in the red and green channel, respectively.

F-actin structures present in SCMs were revealed by STED microscopy and analyzed by plotting changes in fluorescence intensity along line profiles (1–3 μm length) drawn in Image J. Values were exported to Igor Pro 8.0 to quantify periodicity. The average distance between peaks found within an analyzed segment (see for example Fig. 8b–e) was considered as representative of a given neurite.

**Statistics and reproducibility**. Control experiments certifying the action of peptide p4.2[13] were carried out for each culture, considering a culture as a group of SCMs established from a litter of rat pups. The neurons of a given culture were distributed between two or more different experimental protocols. Experimental groups used for evaluating statistical differences passed the Kolmogorov-Smirnov normality test. Data from averages were always expressed as mean ± s.e.m. Comparisons between two groups were established using unpaired two-tailed

Student's *t*-test. When comparing more than two groups one-way ANOVA followed by Bonferroni multiple comparison test was applied. The number of independent observations, as well as the type of statistical test applied and the degree of significance, are indicated in the figures.

**Reporting summary**. Further information on research design is available in the Nature Research Reporting Summary linked to this article.

## Data availability
The data that support the findings of this study are included in the article and supplementary information and, are available from the corresponding author upon reasonable request.

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

## Acknowledgements

This work was sponsored by the Spanish government (Ministerio de Ciencia e Innovación), grant RTI2018-096948-B-100 (A.L.), co-funded by the European Regional Development Fund (ERDF). We also thank CERCA Program/Generalitat de Catalunya for institutional support. A.L. is a Serra Húnter fellow. C.V. was contracted by a MICINN predoctoral fellowship (FPI - BES-2016-076551).

## Author contributions

A.L. conceived the study. C.V. performed experiments and analysed data. A.L. wrote the manuscript.

## Competing interests

The authors declare no competing interests.
