## [Peer Review File · Communications Biology]

Reviewers' comments:

Reviewer #1 (Remarks to the Author):

1. Brief summary of the manuscript

Velasco and Llobet found a biphasic presynaptic response triggered by SPARC C-terminal p4.2 peptide in the autaptic synapses of single cell microculture (SCM) prepared from rat superior cervical ganglion neurons. Electrophysiology revealed that exposure to p4.2 causes synaptic modulation, in which the synaptic strength decreases transiently and then recovers to the original level. The authors presented the data suggesting that the biphasic synaptic response consists of synapse elimination followed by potentiation of presynaptic transmission and de novo synapse formation by performing electrophysiology and immunohistochemistry. Stabilization of F-actin interferes with the initial decline phase of synaptic response, suggesting that synaptic elimination requires reorganization of F-actin. The authors also presented the data that suggest no density changes of postsynaptic receptors in the synaptic elimination phase. Subsequently, calcium imaging and electrophysiology suggest potentiation of presynaptic function occurs in the recovery phase. Based on these data, the authors discussed that the potentiation of presynaptic function is the underlying mechanism of the biphasic synaptic responses caused by exposure to p4.2.

2. Overall evaluation of the work

SPARC secreted from glial cells is of particular interest, since SPARC causes axon retraction leading to behavioral defect in vivo. In the previous work (Lopez-Murcia et al., 2015) from this group, they revealed that SPARC is the trigger for synapse elimination in SCM, and also examined the time course of p4.2 administration on synapse elimination. The effect of the p4.2 administration in both early and late phases was investigated in detail in the current paper, but the identification of underlying molecular mechanism of the late compensation phase is not achieved. Therefore, the conceptual advance of the current work from the previous research should be judged to fall short of the expectations for publication in *Communications Biology*. Also, as described in the specific comments below, several methods need clarification and some of the presented data are not convincing.

3. Specific comments

1. Fig. 1B is not convincing because there is no control (p2.1) that corresponds with p4.2 after 10 h of exposure time. The authors should add the p2.1 data (control) in addition to the p4.2 data after 10 h of exposure time.

2. The EPSC amplitudes at 0 h in Fig. 2E and Fig. 5A are greatly different from that in Fig. 1B. This variability needs explanation.

3. The trend of the data in Fig. 1B, showing recovery already at 30 h, may appear to be inconsistent with the previous data of p4.2 (50 nM @ 48 h, shown in Fig. 3D in Lopez-Murcia et al., 2015), where acceleration of EPSC amplitude decline exists at 48 h. Please explain these two seemingly inconsistent results.

4. In Fig. 2B, bassoon punctum is used as an indicator of a presynapse. Do bassoon puncta truly reflect presynapses? I recommend an experiment to confirm colocalization of bassoon with other presynaptic markers, including synaptobrevin (v-SNARE), synapsin I (synaptic cytoplasmic protein) and synaptophysin (synaptic vesicle protein). Also, it would be necessary to confirm what proportion of bassoon and N-cadherin puncta associated with genuine postsynaptic markers, such as acetylcholine receptors.

5. The correlation coefficient of 0.43 in Fig. 2B and the overall data distribution indicate weak correlation, if exists. The authors should weaken their claim on "linear relationship".

6. In Fig. 4, F0 was calculated from the average images. Could bleaching of GCaMP6f during calcium imaging be negligible? If not, there is a possibility that F0 could not be estimated accurately.

7. In order to increase reliability of the data, I recommend to add data of time lapse images indicating

fluorescence change of GCaMP6 to Fig. 4A.

8. Moreover, it was unclear how the contamination of leak fluorescence signals from the surrounding structures were excluded from fluorescence signals in GCaMP6f(+) puncta. If the leak signals were not excluded, the analysis would provide inaccurate results.

9. In Fig. 4A's legend, the authors described "Discrete, round $\sim 1 \mu\text{m}$ spots that appeared as changes larger than three standard deviations of basal $\Delta F/F_0$ were considered as functional presynaptic terminals". Validity of this threshold should be clarified by adding control experiments in the presence of TTX.

10. In Fig. 4B, F/F_0 in puncta should be correlated to their volumes. Could the classification described here be likely to reflect their volumes?

11. The authors have to add a scale bar to Fig. 6D.

12. Describe the method of determining the cell's longest axis in the data shown in Fig. 7.

13. Describe more details in the interpretation of the data shown in Fig. 7C. Formation of actin bundles without net increase in F-actin leads to both decrease in the distance of nearby single actin filaments and increase in the distance of bundled actin filaments.

14. Adding images of GCaMP6f(+) puncta to Fig. 9 will allow the readers to judge the data quality.

15. The authors have to add a label to the horizontal axis in Fig. 9B.

16. The authors need to describe how background noise in the calcium imaging experiment was determined in the method section.

17. The normality of the data should be checked before performing t-test and this procedure should be stated in the text.

Reviewer #2 (Remarks to the Author):

This manuscript deals with synapse elimination induced by p4.2 peptide, derived from SPARC (a gliasecreted factor that negatively regulates synapse density) in autaptic cultured neurons. Consequently, homeostatic response involving F-actin cytoskeleton reorganization is triggered, concomitant with changes in calcium influx and readily releasable pool size. Findings embedded in this manuscript are potentially important in our understanding of synaptic plasticity. However, I think that coherence in individual figures is not clear and core messages are not presented in a compelling manner.

Major comments:

1. Why did p2.1 treatment for a short time in most of experiments? Moreover, n number of p2.1 in Fig.1 is so low to be put for comparison?

2. Experiment to monitor F-actin remodeling by p4.2 should be performed in neurons to support the authors' claim.

3. Complementary imaging experiments should be performed to validate the physiological observations during synaptic recovery.

4. Current clamping experiments should be performed to examine whether there is any change in intrinsic neuronal property with p4.2 treatment.

5. Why were different interstimulus intervals used in different figures (e.g. Fig.3 and Fig. 8)?

6. More experiments should be done to ensure no postsynaptic changes in autaptic circuits.

Minor comments:

1. Overall, the sentences do not read well. Proofreading is strongly recommended.

2. In line 86 and 279, there are typographical errors.

3. What does "SCM" mean? Full spelling is required when introduced first.

Reviewer #3 (Remarks to the Author):

In the present manuscript, Velasco and Llobet address the role of autaptic signaling for the regulation of synaptic strength. Therefore, they employ a neuronal cell explant model from rat superior cervical ganglion, that was claimed to be free of glial cells and to form autaptic connections. Using a SPARC peptide, a transient decrease of bassoon+ puncta and EPSC were found. These correlated to a transient depression of the readily-releasable pool size and enhanced presynaptic Ca²⁺ influx. While the SPARC peptide under these conditions induced a remodeling of the presynaptic F-actin, impairment of actin dynamics prohibited such compensatory Ca²⁺ signaling. From these data, the conclusion was drawn that autaptic circuits would continuously sense synaptic strength and if that goes down it would lead to compensatory effects that regulate the number of synaptic contacts and neurotransmitter release probability. While the conclusion is generally consistent with the shown data, some ambiguities and missing controls limit the strength of the manuscript in its current state.

Specific points:

1. Throughout the manuscript the authors speak of SCM, but this abbreviation is introduced nowhere. Please amend.
2. In general, the terms autapse and synapse are used in an apparently interchangeable manner. It would, however, be important to better understand, how many of the observed bassoon-puncta are due to synapses from other neurons and how many are autapses. Also, bassoon is used as the only marker for presynapses and none for postsynapses. It would be good to confirm the bassoon data with an additional marker, like piccolo (ideally, at least for some instances, in colocalization) and the postsynapses might be marked by using toxins labeling nicotinic receptors.
3. Fig. 1B: While a time course analysis is provided for the effects of p4.2, the control peptide 2.1 is depicted only for 2 time points. It would be good to see that there is no retarded effect of p2.1.
4. Fig. 7B: to better understand these graphs, indicate the cell borders on top of the actin distribution plots.
5. Text on l. 379f. and Fig. 8D: should the y-axis not be called "decrease in PPR"?
6. In the discussion (l.431-435), the authors speculate about the role of lysosomal transport/activity for the synapse elimination / formation as a response to decreased synaptic strength. It would be good to add some proof to this, e.g. by showing electron micrographs of presynapses / autapses before and after treatment with p4.2 or by following lysotracker signals under these conditions.
7. Overall, language and grammar are fine, some typos / small errors were found in lines: 39, 86, 313, 452.

Reviewer #1:

We thank the reviewer for their time and for their careful consideration of our manuscript.

1. Brief summary of the manuscript. Velasco and Llobet found a biphasic presynaptic response triggered by SPARC C-terminal p4.2 peptide in the autaptic synapses of single cell microculture (SCM) prepared from rat superior cervical ganglion neurons. Electrophysiology revealed that exposure to p4.2 causes synaptic modulation, in which the synaptic strength decreases transiently and then recovers to the original level. The authors presented the data suggesting that the biphasic synaptic response consists of synapse elimination followed by potentiation of presynaptic transmission and de novo synapse formation by performing electrophysiology and immunohistochemistry. Stabilization of F-actin interferes with the initial decline phase of synaptic response, suggesting that synaptic elimination requires reorganization of F-actin. The authors also presented the data that suggest no density changes of postsynaptic receptors in the synaptic elimination phase. Subsequently, calcium imaging and electrophysiology suggest potentiation of presynaptic function occurs in the recovery phase. Based on these data, the authors discussed that the potentiation of presynaptic function is the underlying mechanism of the biphasic synaptic responses caused by exposure to p4.2.

2. Overall evaluation of the work. SPARC secreted from glial cells is of particular interest, since SPARC causes axon retraction leading to behavioral defect in vivo. In the previous work (Lopez-Murcia et al., 2015) from this group, they revealed that SPARC is the trigger for synapse elimination in SCM, and also examined the time course of p4.2 administration on synapse elimination. The effect of the p4.2 administration in both early and late phases was investigated in detail in the current paper, but the identification of underlying molecular mechanism of the late compensation phase is not achieved. Therefore, the conceptual advance of the current work from the previous research should be judged to fall short of the expectations for publication in Communications Biology. Also, as described in the specific comments below, several methods need clarification and some of the presented data are not convincing.

3. Specific comments.

1. Fig. 1B is not convincing because there is no control (p2.1) that corresponds with p4.2 after 10 h of exposure time. The authors should add the p2.1 data (control) in addition to the p4.2 data after 10 h of exposure time.

New experiments have been added to Fig. 1B. Now the figure illustrates that p2.1 did not modify synaptic strength for >24 h. Our previous number of observations (n=24) has increased by three-fold (n=72).

2. The EPSC amplitudes at 0 h in Fig. 2E and Fig. 5A are greatly different from that in Fig. 1B. This variability needs explanation.

Fig. 1B shows the EPSC amplitude of 280 neurons, which came from 50 different cultures recorded over approximately 2 years. All single cell microcultures (SCMs) that

showed neurotransmission were included in this experimental group. We establish microislands by spraying collagen on coverslips, which have a diameter that ranges from 100 to 500 μm . Correlative electrophysiology and immunocytochemistry experiments, as well as electrophysiology experiments combined to SyGCaMP6f were carried out using 63X and 60X objectives, respectively. To obtain a representative imaging of the whole microculture we selected small SCMs (see for example new Fig. 8A). Neurons grown in smaller microcultures tended to display a greater synaptic strength. Moreover, SCMs were grown in culture medium containing 1 to 2 nM CNTF, as stated in the methods section. Experimental groups shown in Fig. 2E and 5A were all treated with the maximal concentration of CNTF, which likely contributed to establish more synapses. The small size of microislands combined with growth in culture medium containing 2 nM CNTF increased basal synaptic strength of the experimental groups shown in Fig. 2E and 5A.

3. The trend of the data in Fig. 1B, showing recovery already at 30 h, may appear to be inconsistent with the previous data of p4.2 (50 nM @ 48 h, shown in Fig. 3D in Lopez-Murcia et al., 2015), where acceleration of EPSC amplitude decline exists at 48 h. Please explain these two seemingly inconsistent results.

The effect of peptide p4.2 is time and concentration dependent. We chose the concentration of 200 nM because it apparently induced a maximal decay of synaptic strength in a time window of 12 hours (Figure S2, Lopez-Murcia et al., 2015). Current data are consistent with this previous observation. In our PNAS paper we thought that synapse elimination was reaching a steady-state value at 12 hours. The present work started by testing this hypothesis and led us to observe the homeostatic compensation for longer incubation times. We do not know if the homeostatic response could also be observed using 50 nM p4.2, after reaching a maximal synapse elimination in a time window of 48 h. It is an interesting question but is far from the scope of the current work.

4. In Fig. 2B, bassoon punctum is used as an indicator of a presynapse. Do bassoon puncta truly reflect presynapses? I recommend an experiment to confirm colocalization of bassoon with other presynaptic markers, including synaptobrevin (v-SNARE), synapsin I (synaptic cytoplasmic protein) and synaptophysin (synaptic vesicle protein). Also, it would be necessary to confirm what proportion of bassoon and N-cadherin puncta associated with genuine postsynaptic markers, such as acetylcholine receptors.

We completely agree with this reviewer. This point should be strengthened. We now show in supplementary Fig.2 that more than 80% of bassoon puncta co-localized with synapsin-I and VAMP-2. Staining with piccolo (Abcam, ab20664), a presynaptic active zone marker, was unsuccessful.

We have not managed to get good co-stainings with postsynaptic markers. We first tried to label acetylcholine receptors with α -bungarotoxin-Alexa 555 (Life technologies, B35451) but, stainings were unsuccessful because acetylcholine receptors in autonomic ganglia contain a combination of α 3 and β 4 subunits and lack α 7 subunits. So, we next tested a polyclonal antibody against α 3 subunits (Abcam, ab183097) to combine it with the anti-bassoon monoclonal used for stainings (Enzo Life Sciences, ADI-VAM-PS003). Labelling was completely unspecific. As an alternative we tested a

polyclonal antibody against PSD-93 (Abcam, ab2930). Again, stainings were unsuccessful.

Maximum intensity projections of three different SCMs stained for piccolo (1:1000), PSD-93 (1:1000) and α -bungarotoxin ($1\mu\text{g}/\text{mL}$, 2h of incubation).

In summary, we can only provide images of double labelling of bassoon with 2 different presynaptic markers (Figure S2), which reinforces the identity of bassoon puncta as synapses.

5. The correlation coefficient of 0.43 in Fig. 2B and the overall data distribution indicate weak correlation, if exists. The authors should weaken their claim on “linear relationship”.

The reviewer is right. We have now strengthened our claims by increasing the number of observations from 32 to 41. As a result, the Pearson correlation coefficient of the linear fit in Fig. 2B has increased to 0.54 (Figure 2 legend). We think that this result supports that a linear function described reasonably well the association between EPSC amplitude and bassoon puncta. The variability in individual release probability clearly limited the goodness of the fit.

6. In Fig. 4, F_0 was calculated from the average images. Could bleaching of GCaMP6f during calcium imaging be negligible? If not, there is a possibility that F_0 could not be estimated accurately.

We have re-written and expanded the methods section explaining SyGCaMP6f measurements (lines 412-424). The contribution of photobleaching was negligible. Following the suggestion of this reviewer we used TTX to block neurotransmission. New Figure 4B shows that SyGCaMP6f fluorescence was stable during the recording period when synapses did not respond, thus only the effect of photobleaching could be seen.

7. In order to increase reliability of the data, I recommend to add data of time lapse images indicating fluorescence change of GCaMP6 to Fig. 4A.

Supplementary movie 1 now shows the time course of changes in SyGCaMP6f fluorescence displayed in Fig.4A.

8. Moreover, it was unclear how the contamination of leak fluorescence signals from the surrounding structures were excluded from fluorescence signals in GCaMP6f(+) puncta. If the leak signals were not excluded, the analysis would provide inaccurate results.

Fluorescence generated outside of the microculture was subtracted at the beginning of analysis. The contamination of signals arising from synapses out of focus was minimized by considering as synapses only those puncta that showed increases in $\Delta F/F_0$ larger than 3 standard deviations in baseline values. New Fig. 4B shows the individual responses that were above the threshold set by 3 SDs in the baseline period (see also below).

9. In Fig. 4A's legend, the authors described "Discrete, round $\sim 1 \mu\text{m}$ spots that appeared as changes larger than three standard deviations of basal $\Delta F/F_0$ were considered as functional presynaptic terminals". Validity of this threshold should be clarified by adding control experiments in the presence of TTX.

Thank you for the suggestion. Figure 4B now shows the response of a neuron before and after exposure to $1 \mu\text{M}$ TTX.

10. In Fig. 4B, F/F_0 in puncta should be correlated to their volumes. Could the classification described here be likely to reflect their volumes?

We identified putative functional synapses in difference images, which detected the absolute changes in fluorescence achieved during stimulation. This procedure allowed us to identify the center of mass of all puncta susceptible of being a synapse. We next draw $1.3 \times 1.3 \mu\text{m}$ ROIs and considered as functional presynaptic terminals those that experimented increases in $\Delta F/F_0$ larger than 3 standard deviations in baseline values. This criterion was useful to eliminate fluorescence emanating from synapses located out of focus. Considering that a typical presynaptic terminal has a diameter of $1 \mu\text{m}$ and the optical resolution of our 60X/1.25 NA objective was found between 250-300 nm, we can be reasonably confident that we were isolating individual synapses. However, our system could not resolve differences in presynaptic terminal size, i.e. from 0.8 to $1.2 \mu\text{m}$ diameter, therefore it is unlikely that the changes detected in $\Delta F/F_0$ were related to variations in presynaptic terminal volume.

11. The authors have to add a scale bar to Fig. 6D.

Thanks. Corrected.

12. Describe the method of determining the cell's longest axis in the data shown in Fig.7.

We used Feret's diameter implemented in Image J. Now detailed in line 450.

13. Describe more details in the interpretation of the data shown in Fig. 7C. Formation of actin bundles without net increase in F-actin leads to both decrease in the distance of nearby single actin filaments and increase in the distance of bundled actin filaments.

We have made an effort to strengthen the part of the manuscript related to the effects of p4.2 on F-actin. Experiments carried out in cell lines show that this peptide induces a redistribution of filamentous actin in the cell without inducing its depolymerization (Fig. 7). The ability of p4.2 to reorganize F-actin has now been evaluated in neurons

using STED microscopy. The peptide disrupts acting ring-like structures found in neuronal processes, which could be a trigger for synapse elimination. This new set of data are now illustrated in new Fig. 8 (lines 203-210).

14. Adding images of GCaMP6f(+) puncta to Fig. 9 will allow the readers to judge the data quality.

The previous Fig.9 is now Fig. 10. Panel A shows 4 difference images in the conditions relevant to the figure. The goal is to show the number of putative synaptic puncta present in a SCM.

15. The authors have to add a label to the horizontal axis in Fig. 9B.

Sorry about this. Corrected.

16. The authors need to describe how background noise in the calcium imaging experiment was determined in the method section.

Background fluorescence was estimated from ROIs located outside of the microculture (lines 413-416).

17. The normality of the data should be checked before performing t-test and this procedure should be stated in the text.

Data passed the Kolmogorov-Smirnov test before applying parametric tests. Now stated in the methods section (line 467).

Reviewer #2 (Remarks to the Author):

We thank the reviewer for their time and for their careful consideration of our manuscript.

This manuscript deals with synapse elimination induced by p4.2 peptide, derived from SPARC (a glia-secreted factor that negatively regulates synapse density) in autaptic cultured neurons. Consequently, homeostatic response involving F-actin cytoskeleton reorganization is triggered, concomitant with changes in calcium influx and readily releasable pool size. Findings embedded in this manuscript are potentially important in our understanding of synaptic plasticity. However, I think that coherence in individual figures is not clear and core messages are not presented in a compelling manner.

Major comments:

1. Why did p2.1 treatment for a short time in most of experiments? Moreover, n number of p2.1 in Fig.1 is so low to be put for comparison?

We agree with this reviewer. The number of observations in Fig.1B related to peptide p2.1 have been increased from 24 to 72 to allow a better comparison with peptide p4.2. Similarly, the effect of peptide 2.1 in Fig. 6C is now shown for periods longer than 20 h (8 more cells added).

2. Experiment to monitor F-actin remodeling by p4.2 should be performed in neurons to support the authors' claim.

This comment led us to carry out STED experiments to visualize periodic actin structures in neuronal processes. Control neurons showed actin ring-like structures at a regular spacing of 195 nm. This value is comparable to distances found in dendrites and axons of cultured hippocampal neurons (see for example Bar et al., Sci Rep, 2016; Han et al, PNAS 2017). Exposure to p4.2 disrupted this characteristic organization and although a certain periodicity was observed in F-actin structures, a ring-like organization was absent. This result reinforces our findings and suggests that the action of peptide p4.2 might come from a reorganization of the neuronal F-actin skeleton. Lines 203-210.

3. Complementary imaging experiments should be performed to validate the physiological observations during synaptic recovery.

STED microscopy shows that actin ring-like structures were not present by the end of the recovery phase. This result indicates that compensation overcomes the alterations in the actin cytoskeleton and strengthens our discussion (specifically, lines 296-302).

4. Current clamping experiments should be performed to examine whether there is any change in intrinsic neuronal property with p4.2 treatment.

This experiment is now included in supplementary Fig. 1. Synapse elimination occurred without a significant alteration in neuronal excitability.

5. Why were different interstimulus intervals used in different figures (e.g. Fig.3 and Fig. 8)?

We analyzed the effect of p4.2 exposure on short-term plasticity evaluating paired-pulse ratio. We observed a general tendency to decrease PPR at the end of the

recovery phase but, a significant difference was only observed for a 50 ms interval. This effect is illustrated in Fig 3. In old figure 8 (now Fig. 9) we did not use the time interval of 50 ms and chose 100 ms instead. Fig. 9C-top shows that paired pulse depression for this time interval decreased at the end of the recovery phase, albeit not significantly. Therefore, the longer was the interval between stimuli, the less obvious was the decrease in paired-pulse depression at the end of the homeostatic response. We took advantage of a non-significant difference in paired pulse depression for a time interval of 100 ms to see the effects of latrunculin-A on PPR (Fig. 9C-bottom).

6. More experiments should be done to ensure no postsynaptic changes in autaptic circuits.

We have added more experiments evaluating the contribution of postsynaptic receptors. New experiments have been included in Fig. 6B to show the effect of electrical and chemical stimulations over exposure to peptide p4.2 for 30 hours. The number of experiments has been increased from 42 to 60 independent observations. Fig. 6C now shows that peptide p2.1 does not affect EPSC amplitude or nicotinic currents.

Minor comments:

1. Overall, the sentences do not read well. Proofreading is strongly recommended.

We have carefully reviewed the manuscript to facilitate reading.

2. In line 86 and 279, there are typographical errors.

Thanks. Corrected.

3. What does "SCM" mean? Full spelling is required when introduced first.

Sorry about this. It means Single Cell Microculture (SCM), because a single neuron establishes a microculture in the absence of any other cell type. It is now clearly stated in line 61.

Reviewer #3 (Remarks to the Author):

We thank the reviewer for their time and for their careful consideration of our manuscript.

In the present manuscript, Velasco and Lobet address the role of autaptic signaling for the regulation of synaptic strength. Therefore, they employ a neuronal cell explant model from rat superior cervical ganglion, that was claimed to be free of glial cells and to form autaptic connections. Using a SPARC peptide, a transient decrease of bassoon+ puncta and EPSC were found. These correlated to a transient depression of the readily-releasable pool size and enhanced presynaptic Ca²⁺ influx. While the SPARC peptide under these conditions induced a remodeling of the presynaptic F-actin, impairment of actin dynamics prohibited such compensatory Ca²⁺ signaling. From these data, the conclusion was drawn that autaptic circuits would continuously sense synaptic strength and if that goes down it would lead to compensatory effects that regulate the number of synaptic contacts and neurotransmitter release probability. While the conclusion is generally consistent with the shown data, some ambiguities and missing controls limit the strength of the manuscript in its current state. Specific points:

1. Throughout the manuscript the authors speak of SCM, but this abbreviation is introduced nowhere. Please amend.

Sorry about this. The term SCM stands for Single Cell Microculture (SCM), because a single neuron establishes a microculture in the absence of any other cell type. It is now clearly stated in line 61 at the beginning of the results section.

2. In general, the terms autapse and synapse are used in an apparently interchangeable manner. It would, however, be important to better understand, how many of the observed bassoon-puncta are due to synapses from other neurons and how many are autapses. Also, bassoon is used as the only marker for presynapses and none for postsynapses. It would be good to confirm the bassoon data with an additional marker, like piccolo (ideally, at least for some instances, in colocalization) and the postsynapses might be marked by using toxins labeling nicotinic receptors.

Sorry for the lack of clarity. All synapses present in our cultures were autapses. The paper now includes new supplementary figure 2 to show that: i) SCMs contained a single nucleus (labelled with DAPI) and ii) bassoon co-localized with the presynaptic markers VAMP-2 and synapsin-I. Following the suggestions of this reviewer we tried a double immunostaining for bassoon (Enzo Life Sciences, ADI-VAM-PS003) and piccolo (Abcam, ab20664). The piccolo antibody did not work and provided a completely unspecific labelling. We also tried to label acetylcholine receptors with α -bungarotoxin-Alexa 555 (Life technologies, B35451) but stainings were unsuccessful because acetylcholine receptors in autonomic ganglia contain a combination of α 3 and β 4 subunits and lack α 7 subunits. So, we used a polyclonal antibody against α -3 subunits (Abcam, ab183097). Labelling was completely unspecific, again. Finally, we tested a polyclonal antibody against PSD-93 (Abcam, ab2930) and labelling was unspecific.

Maximum intensity projections of three different SCMs stained for piccolo (1:1000), PSD-93 (1:1000) and α -bungarotoxin (1 μ g/ml, 2h of incubation).

3. Fig. 1B: While a time course analysis is provided for the effects of p4.2, the control peptide 2.1 is depicted only for 2 time points. It would be good to see that there is no retarded effect of p2.1.

This is definitely a good point. We have added new experiments evaluating the effect of peptide p2.1. The number of observations in Fig.1B have been increased from 24 to 72 to allow a better comparison with peptide p4.2. Similarly, the effect of peptide 2.1 in Fig. 6C is now shown for periods longer than 20 h (8 more cells added).

4. Fig. 7B: to better understand these graphs, indicate the cell borders on top of the actin distribution plots.

Corrected.

5. Text on l. 379f. and Fig. 8D: should the y-axis not be called "decrease in PPR"?

We agree, the figure was not clear enough. Fig.8 is now Fig.9 because we have expanded the part of the paper related to modifications in F-actin. Fig. 8 shows STED microscopy images of F-actin stainings. The peptide disrupts acting ring-like structures found in neuronal processes (lines 203-210). The term "decrease in PPR" was confusing and made reference to the average PPR found from 7.5- to 10 min of latrunculin-A exposure. Axis label has been modified. The time constant of the exponential fit aims to provide a quantification of the time course of changes occurring in the F-actin cytoskeleton. (lines 763-767).

6. In the discussion (l.431-435), the authors speculate about the role of lysosomal transport/activity for the synapse elimination / formation as a response to decreased synaptic strength. It would be good to add some proof to this, e.g. by showing electron micrographs of presynapses / autapses before and after treatment with p4.2 or by following lysotracker signals under these conditions.

This part of the discussion was based on previous data, please see Fig.S3, Fig. S4 and Fig.4 in Lopez-Murcia et al., PNAS 2015. We are confident that lysosomes are important for the cell-autonomous mechanism of synapse elimination induced by peptide p4.2. The discussion brings the idea that the arrival of lysosomal compartments digesting presynaptic elements could be relevant for orchestrating the

homeostatic response. We have no evidence for this at the moment, however, our new findings showing that peptide p4.2 disrupts periodic F-actin repeats along neuronal processes can provide an explanation of how peptide p4.2 de-stabilizes synaptic contacts leading to a lysosomal-mediated digestion. This view is now included in the discussion (lines 269-274).

7. Overall, language and grammar are fine, some typos / small errors were found in lines: 39, 86, 313, 452.

Thanks. Corrected.

REVIEWERS' COMMENTS:

Reviewer #2 (Remarks to the Author):

The authors addressed most of comments in a satisfactory manner, which I think significantly enhanced the quality of the manuscript. I recommend the publication of the revised manuscript.

Reviewer #3 (Remarks to the Author):

My concerns were properly addressed.